# VirBR, a transcription regulator, promotes IncX3 plasmid transmission, and persistence of *bla*NDM-5 in zoonotic bacteria

Tengfei Ma[1,6], Ning Xie[1,6], Yuan Gao[1], Jiani Fu[1], Chun E. Tan[1], Qiu E. Yang[2], Shaolin Wang[1], Zhangqi Shen [1], Quanjiang Ji [3], Julian Parkhill [4], Congming Wu[1], Yang Wang [1] ✉, Timothy R. Walsh[5] ✉ & Jianzhong Shen [1] ✉

IncX3 plasmids carrying the New Delhi metallo-β-lactamase-encoding gene, *bla*NDM-5, are rapidly spreading globally in both humans and animals. Given that carbapenems are listed on the WHO AWaRe watch group and are prohibited for use in animals, the drivers for the successful dissemination of Carbapenem-Resistant Enterobacterales (CRE) carrying *bla*NDM-5-IncX3 plasmids still remain unknown. We observe that *E. coli* carrying *bla*NDM-5-IncX3 can persist in chicken intestines either under the administration of amoxicillin, one of the largest veterinary β-lactams used in livestock, or without any antibiotic pressure. We therefore characterise the *bla*NDM-5-IncX3 plasmid and identify a transcription regulator, VirBR, that binds to the promoter of the regulator gene *actX* enhancing the transcription of Type IV secretion systems (T4SS); thereby, promoting conjugation of IncX3 plasmids, increasing pili adhesion capacity and enhancing the colonisation of *bla*NDM-5-IncX3 transconjugants in animal digestive tracts. Our mechanistic and in-vivo studies identify VirBR as a major factor in the successful spread of *bla*NDM-5-IncX3 across one-health AMR sectors. Furthermore, VirBR enhances the plasmid conjugation and T4SS expression by the presence of copper and zinc ions, thereby having profound ramifications on the use of universal animal feeds.

The emergence and prevalence of carbapenem-resistant Enterobacterales (CRE) are regarded as a global health threat. CRE causes invasive infections in health-care settings commonly associated with high mortality, and options for treating CRE infections are becoming increasingly limited[1–4]. The most common mechanism of CRE is the expression of ß-lactam-hydrolysing enzymes e.g. New Delhi metallo-β-lactamase (NDM) and *Klebsiella pneumoniae* carbapenemase (KPC), encoded by *bla*NDM and *bla*KPC, respectively[5,6]. Unlike *bla*KPC mainly found in bacteria from human infections; *bla*NDM can be found across both humans and animal sectors[7–10]. Since the first *bla*NDM variant, *bla*NDM-1, was identified in *Escherichia coli* and *K. pneumoniae* from a patient in India in 2008[11], 41 additional variants (*bla*NDM-2 to *bla*NDM-42) have been identified mainly in Enterobacterales. Among these many *bla*NDM variants, *bla*NDM-5-positive *E. coli* from both human and animal origin showed an increasing trend from 17 countries in 2009–2014 to 34 countries in 2015–2022 (Fig. S1a, b). 73.4% of 3203 NDM-producing *E. coli* isolates from animals and humans in 2009–2022 collected from GenBank database (https://www.ncbi.nlm.nih.gov/genbank/) were shown to be *bla*NDM-5-positive (Fig. S2).

[1]National Key Laboratory of Veterinary Public Health and Safety, College of Veterinary Medicine, China Agricultural University, Beijing, China. [2]College of Environment and Resources, Fujian Agriculture and Forestry University, Fuzhou, China. [3]School of Physical Science and Technology, Shanghai Tech University, Shanghai, China. [4]Department of Veterinary Medicine, University of Cambridge, Cambridge, UK. [5]Ineos Oxford Institute for Antimicrobial Research, Department of Biology, Oxford, UK. [6]These authors contributed equally: Tengfei Ma, Ning Xie. ✉e-mail: wangyang@cau.edu.cn; timothy.walsh@biology.ox.ac.uk; sjz@cau.edu.cn

Plasmids play a critical role in the global spread of antimicrobial resistance genes; not least, IncX3-type plasmids are largely associated with $bla_{NDM-5}$ in *E. coli* isolated from human infections, human faecal carriage, and food-producing animals[7,8,12,13]. IncX3 is a member of the family of IncX-type plasmids, which pre-dates the antibiotic era and are generally considered narrow-host-range isolated mainly from Enterobacterales[14]. The earliest record of an *E. coli* carrying IncX3 (designated pEC14_35), was from the United States in 1989, and was approx. 35 kb in size but absent of any antibiotic resistance genes (ARGs) (Fig. S3a, b). Subsequently, IncX3 plasmids have become globally prominent and have acquired many ARGs including $bla_{CTX-M}$, $bla_{TEM}$, $bla_{SHV}$, $bla_{OXA}$, and $bla_{KPC}$ (Figs. S3ab and S4). Genomic mining data suggests that in 2010, $bla_{NDM-1}$-carrying insertion sequences and transposable elements became embedded in IncX3, and $bla_{NDM-5}$ was first associated with IncX3 in 2015. Currently, most $bla_{NDM-5}$-IncX3 plasmids are invariably reported with a size of ~46 kb (Fig. S3b).

It should be noted that the use of carbapenems is prohibited in food-producing animals and is only administered via IV/IM for human clinical infections; therefore, it is counter-intuitive that $bla_{NDM-5}$-IncX3 plasmids are so prevalent in animal and human faeces[7,8,12]. Herein we proposed two hypotheses: (1) Veterinary-approved β-lactams may promote the colonisation and expansion of $bla_{NDM-5}$-IncX3 strains in animal gastro-intestinal tracts; (2) Unidentified genes located on $bla_{NDM-5}$-IncX3 might play a role in aiding the plasmid's persistence in *E. coli* while in-situ.

Accordingly, in this work we demonstrate that amoxicillin, an extensively used β-lactam in animals, can promote the persistence of $bla_{NDM-5}$-IncX3 *E. coli* strains in chicken gastro-intestinal tracts.

Specifically, we identify a transcription regulator gene *virBR*, encoding VirBR that can bind the promoter region of a regulator gene, *actX*, which enhances the transcription of type IV secretion systems (T4SS), and augments the conjugation of IncX3 plasmids, thereby, successfully promoting host gastro-intestinal tract colonisation and dissemination even without antibiotic pressure. However, we further show that the presence of copper and zinc activates the expression of VirBR-mediated T4SS. Copper and zinc are extensively used in farming and may play an essential role in driving the spreading of IncX3 plasmids.

## Results

### Persistence of $bla_{NDM-5}$-positive *E. coli* and transfer of $bla_{NDM-5}$-IncX3 plasmids under in-vivo conditions

To explore the trends of $bla_{NDM-5}$-positive *E. coli* in the chicken gastro-intestinal tract, we inoculated an *E. coli* strain, named 3R, carrying $bla_{NDM-5}$-IncX3 plasmid isolated from chicken in a commercial farm[8], into NDM-free chickens with and without amoxicillin treatment (Fig. 1a). The level of $bla_{NDM-5}$-positive *E. coli* in 3R group treated with amoxicillin from 6 to 40 days remained at approximately $10^8$-$10^{10}$ colony-forming units (CFU)/g at each sampling point (Fig. 1b). Notably, the inoculated $bla_{NDM-5}$-positive *E. coli* in 3R group without antibiotic treatment persisted in the broiler cecum at $10^3$-$10^6$ CFU/g from 8 to 47 days (Fig. 1b). Additionally, under the selective pressure of amoxicillin, $bla_{NDM-5}$ transferred from 3R into intestinal commensal strains after 2 days post-inoculation (10-day-old chickens), and the number of transconjugants remained stable and high at ~$1.0 \times 10^8$ CFU/g (Fig. 1b). Meanwhile, we also observed that amoxicillin increased the in vitro

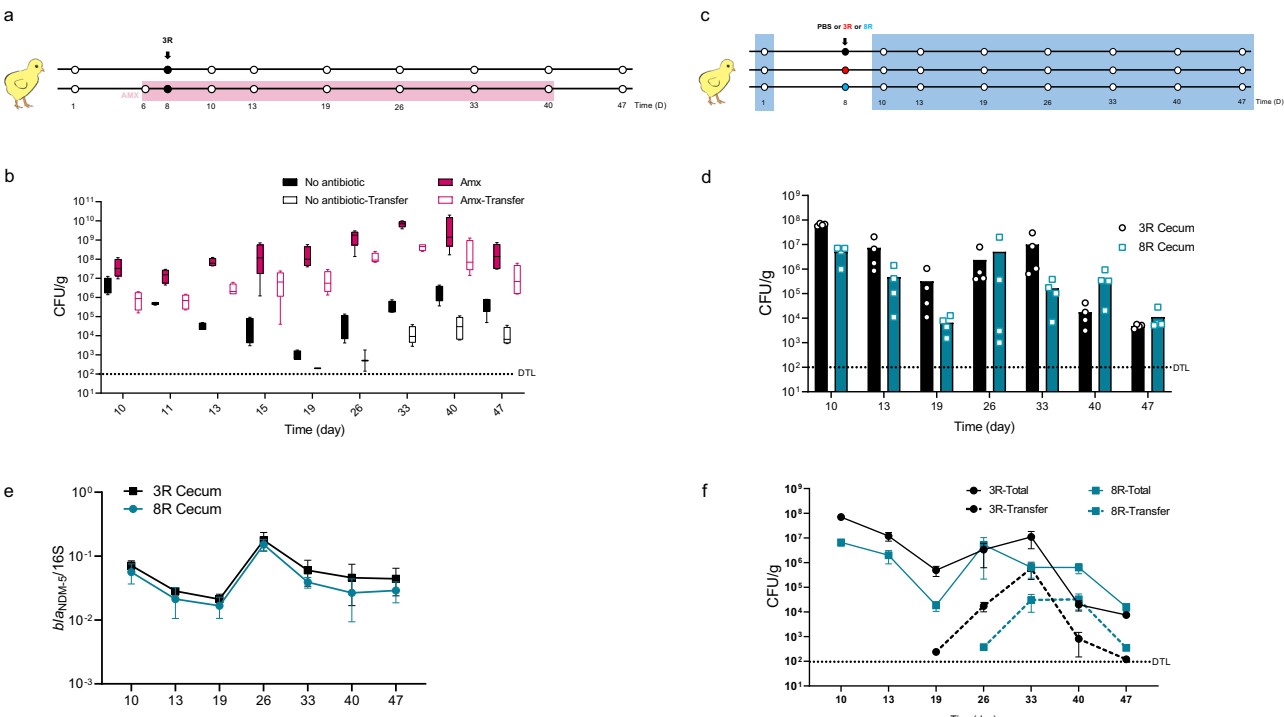

**Fig. 1 | In-vivo stability of $bla_{NDM-5}$-IncX3 plasmid positive *E. coli* 3R and 8R and transfer of $bla_{NDM-5}$-IncX3 to microbiota commensal strains. a** Outline of the experimental design illustrating time (days) of inoculum and sample collection of 3R and 3R plus amoxicillin. Black circle indicated strain 3R and white circle indicated time of sampling cecum. The pink highlighted line indicates that chickens in this group were not only orally administered the 3R strains but also treated with amoxicillin. **b** Cecum abundance (cfu/g) over time (days). Black boxes indicated no antibiotic and red boxes indicated the addition of amoxicillin. "Transfer" with white box indicated transfer of the $bla_{NDM-5}$ gene into chicken commensal strains. DTL indicated the detection line. Lines are medians, boxes cover the 25th to 75th percentiles, and whiskers show the range; $n = 4$ biologically independent replicates. **c** Outline of experimental design illustrating time (days) of inoculum and sample collection of placebo group (white circle), 3R (black circle), and 8R (blue circle). **d** Cecum abundance (cfu/g) of 3R (black column) and 8R (blue column) over time (days). $n = 4$ biologically independent replicates examined over three independent experiments. **e** Abundance of $bla_{NDM}$ genes in cecum of 3R (black line) and 8R (blue line) strains without amoxicillin treatment over time (days). Data are means ± SEM; $n = 4$ biologically independent replicates. **f** Abundance of $bla_{NDM-5}$-positive total strains without amoxicillin treatment over time (days). Dotted lines represent the abundance of transfer of plasmids from 3R (black line) and 8R (blue line) to cecum commensal strains. Data are means ± SEM; $n = 4$ biologically independent replicates. Source data are provided as a Source Data file.

conjugative transfer efficiency of the $bla_{NDM-5}$-IncX3 plasmid ($p < 0.05$; Fig. S5). Intriguingly, *E. coli* 3R carries chromosome-borne $bla_{CTX-M-65}$, which has a lower copy number than the plasmid-borne $bla_{NDM-5}$ and produces the CTX-M-65 enzyme. This ß-lactam-hydrolysing enzyme exhibits lower *Kcat* and *Kcat*/*Km* values for aminopenicillins compared to NDM-5[15,16]. Despite this, amoxicillin was observed to promote the conjugative transfer of $bla_{NDM-5}$-IncX3 plasmid in vitro, even in the absence of $bla_{CTX-M-65}$ in *E. coli* 3R (Fig. S5). In addition, without antibiotic pressure, the $bla_{NDM-5}$-positive transconjugants were isolated after 11 days post-inoculation (19-day-old chickens), with frequencies ranging $10^1$-$10^4$ CFU/g (Fig. 1b).

To confirm that the gastro-intestinal tract persistence of $bla_{NDM-5}$-positive bacteria is not host-dependent, we repeated the in-vivo study by substituting *E. coli* 3R with another $bla_{NDM-5}$-IncX3-positive *E. coli* strain, 8R (also originally isolated from a commercial chicken farm[8]). Similar to 3R, *E. coli* 8R was fed to $bla_{NDM-5}$-IncX3-free chickens without amoxicillin (Fig. 1c). For the negative control group, no $bla_{NDM}$-positive isolates or $bla_{NDM-5}$ were detected throughout the breeding period (1–47 days), whilst both 8R and 3R strains persisted in all parts of intestines (Figs. 1d and S6a, b), although bacterial loads in the large intestines were higher than those in the small intestines ($10^4$-$10^7$ versus $10^2$-$10^3$ CFU/g, $p < 0.001$; Fig. S6c, d). Moreover, in both 8R and 3R groups without amoxicillin treatment, the presence of the $bla_{NDM}$ gene remained stable (Fig. 1e). $bla_{NDM-5}$-positive in-vivo transconjugants, mainly *E. coli* of various ST types and two *K. pneumoniae* strains, emerged in the chickens at 19- and 26-day-old of 3R and 8R groups without amoxicillin treatment, respectively (Figs. 1f and S7a, b). The intact $bla_{NDM-5}$-IncX3 plasmid was identified in all 27 randomly selected *E. coli* transconjugants (Fig. S7c). Thus, amoxicillin promotes the transfer of $bla_{NDM-5}$-IncX3 both in vivo and in vitro; furthermore, $bla_{NDM-5}$-IncX3-positive *E. coli* 3R and 8R persisted and remained stable in chicken intestines without antibiotic pressure.

## Deletion of *virBR* gene on $bla_{NDM-5}$-IncX3 plasmid decreases *E. coli* persistence

As the intestinal persistence of $bla_{NDM-5}$-positive *E. coli* strains is not singularly associated with amoxicillin pressure, we propose that the $bla_{NDM-5}$-IncX3 plasmid may possess other functions that augment the persistence of *E. coli* in the chicken gastro-intestinal tract. As *tra*-T4SS, a homolog of the prototypical VirB/VirD4 system in *Agrobacterium tumefaciens*, was identified to be a key factor for the colonisation and persistence of *E. coli* in the gastro-intestinal tract, we explored whether a plasmid-borne T4SS presented on IncX3 and other IncX-type plasmids promotes bacterial persistence (Fig. S8a,b). Accordingly, we found a highly conserved region including a T4SS cluster (-15 kb) and a corresponding upstream region (-1 kb) in IncX1, IncX2, IncX3 and IncX5-type plasmids. Following deletion of one of these unknown genes, designated *virBR* (3R-Δ*virBR*), then compared the conjugation ability of 3R and 3R-Δ*virBR* into *E. coli* strain J53[17]. We observed the conjugation frequency of $bla_{NDM-5}$-IncX3 was significantly decreased from $3.2 \times 10^{-3}$ in strain 3R to $6.4 \times 10^{-5}$ (Fig. 2a). Simultaneously, we also transferred the intact $bla_{NDM-5}$-IncX3 plasmid into the standard *E. coli* strain BW25113[18] (BW25113-IncX3) and then knocked out the *virBR* gene (BW25113-IncX3-Δ*virBR*), where the conjugation frequency showed a similar declining trend as 3R, decreased from $9.4 \times 10^{-1}$ to $8.2 \times 10^{-4}$ (Fig. S9). Moreover, we continued monitoring and comparing the invasion capabilities of the intact IncX3 plasmid and the IncX3-Δ*virBR* plasmid on the population of plasmid-free *E. coli* BW25113 for 5 days. The results indicated that the intact IncX3 plasmid successfully invaded into plasmid-free *E. coli* BW25113 population after 2 days when BW25113-IncX3 and BW25113 were co-cultured (Fig. S10a), whilst the IncX3-Δ*virBR* plasmid failed to invade into plasmid-free BW25113 population even after 5 days of co-culture (Fig. S10b). Furthermore, IncX3 plasmid, not IncX3-Δ*virBR* plasmid, occupied BW25113 after 3 days when BW25113, BW25113-IncX3 and BW25113-IncX3-Δ*virBR* were co-cultured (Fig. S10c).

To further evaluate the impact of plasmid-borne *virBR* on its host *E. coli*, chicken studies were undertaken and revealed that the colonisation ability of $bla_{NDM-5}$-IncX3-positive *E. coli* were impaired after deleting *virBR* (Fig. S11a, b). Compared with 3R *E. coli* strains, the 3R-Δ*virBR* *E. coli* in the chicken gastro-intestinal tract were significantly decreased from $10^6$ CFU/g to less than $10^2$ CFU/g (DTL, detection limit) in 3 days (Fig. 2b). Furthermore, when the colonisation studies were repeated in a murine model, we showed that 3R-Δ*virBR* *E. coli* strains in mice decreased significantly after 14 days of inoculation (from $10^{10}$ CFU/g to $10^4$ CFU/g) and could not be detected after 20 days post inoculation ($<10^2$ CFU/g) (Fig. 2c). Given the differences seen between BW25113-IncX3-Δ*virBR* and BW25113-IncX3, in-vitro cell line analysis was undertaken to examine the effect of VirBR on *E. coli* adherence and other cellular properties. BW25113-IncX3-Δ*virBR* *E. coli* strains exhibited a weakened auto-aggregation ability that is commonly associated with bacterial adhesion to cell monolayers[19], and the complementation of *virBR* significantly recovered this ability (Fig. S10d). The cell lines used were human colon adenocarcinoma cells, Caco-2, and rat small intestine cell IEC-6 cells, to evaluate the adhesion ability of 3R and 3R-Δ*virBR* strains. The adhesion data show that only 50% and 25% 3R-Δ*virBR* adhered to Caco-2 and IEC-6 cells, respectively (Fig. 2d, e). We subsequently visualised the *E. coli* 3R-Δ*virBR* pili using negative staining transmission electron microscopy (TEM) to understand possible differences in adhesion ability, and observed that the pili structures of *E. coli* 3R-Δ*virBR* are appreciably diminished compared with the intact *E. coli* 3R strain (Fig. 2f). Collectively, these results indicate that *virBR* carried is associated with enhanced colonisation of $bla_{NDM-5}$-IncX3-positive *E. coli*.

## VirBR promotes the transcription of T4SS by enhancing the activity of *actX*

To explore the mechanisms for the decrease in plasmid conjugation frequency and strain colonisation with Δ*virBR*, RNA-sequencing was conducted to examine any possible regulatory functions associated with VirBR. When *virBR* was overexpressed, 38 genes were up-regulated and four genes were down-regulated (Fig. S12a). Most of the up-regulated genes (63.16%, $n = 24$) were present on $bla_{NDM-5}$-IncX3 (Fig. S12a). Among which, 22 genes including the conjugation-related T4SS genes *virB1-11/D4* ($n = 10$), were located downstream of *actX*, an essential regulator gene for T4SS (Fig. 3a). The up-regulated T4SS might mediate the mating-induced SOS response[20], as we observed eight SOS-response related genes, five located on the chromosome and three in plasmids (other than IncX3) that were up-regulated (Fig. S12b). The remaining six up-regulated genes and four down-regulated genes were located on the chromosome (Fig. S12a, Supplementary Table 1 and Supplementary Table 2). RT-qPCR was deployed to confirm that the transcripts of T4SS-related genes decreased with Δ*virBR* and increased after overexpressing *virBR* (Fig. S13). We therefore hypothesized that *virBR* may promote the transcription of T4SS related genes via the promoter activity of *actX*. Accordingly, a GFP fluorescence reporter gene was fused under the control of promoter regions of three genes located downstream of *virBR*. These were: (1) relaxation enzyme-encoding gene *taxA*, (2) *actX* and (3) conjugation transfer-associated gene *virB1* (Fig. S14a). The promoter activity of *actX*, rather than *taxA* and *virB1*, was significantly increased by 3.5-fold when coexisting with *virBR*, indicating that *virBR* enhances the promoter activity of *actX* (Fig. S14b). Additionally, our RT-PCR data confirms the co-transcription profile of *actX* and *virB1* (Fig. S15), further indicating that *virBR* indirectly enhances the transcription of *virB1-11/D4* gene clusters by increasing the activation of *actX*. To explore possible exogenous signals regulating VirBR, we examined temperature, pH, nutrient availability, and heavy metals ($Cu^{2+}$ and $Zn^{2+}$). We observed the elevated conjugation frequency (2-3

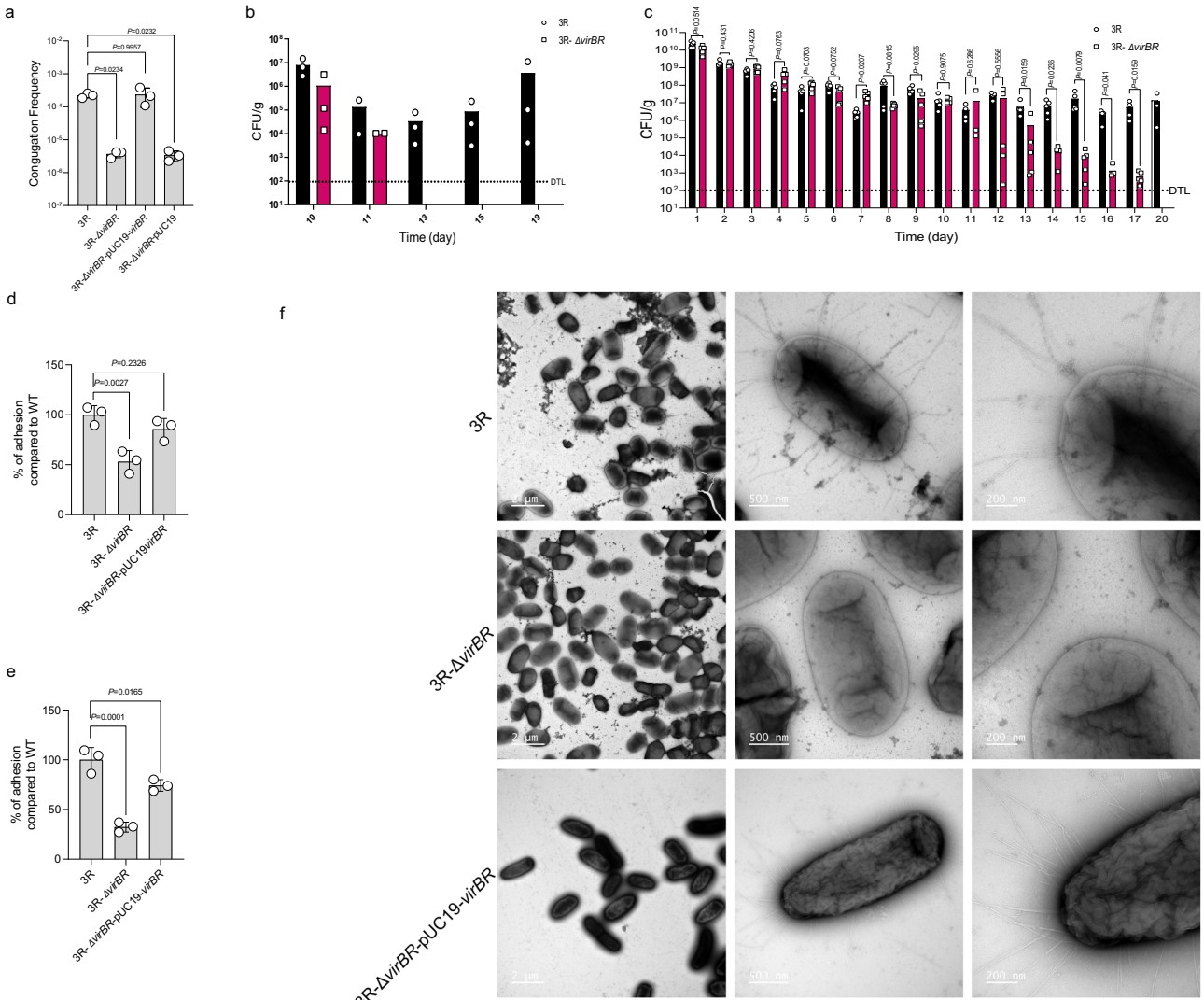

**Fig. 2 | The conjugation ability of IncX3 plasmid, and the effect of ΔvirBR on colonisation, adhesion and pili formation. a** The in-vitro conjugation frequency of 3R, 3R-ΔvirBR, 3R-ΔvirBR-pUC19-virBR, 3R-ΔvirBR-pUC19 strains. Data are means ± SEM; n = 3 biologically independent replicates obtained from three independent experiments. One-way ANOVA and Tukey's multiple comparisons test were performed on values. **b** Abundance in chicken cecum of 3R (circles and black bars) and 3R-ΔvirBR group (squares and red bars) over time (days). n = 3 biologically independent replicates. **c** Abundance in mice faeces of 3R (circles and black bars) and 3R-ΔvirBR group (squares and red bars) over time (days). The detail experimental design can be found in Fig. S10. n ≥ 3 biologically independent replicates obtained from three independent experiments. Groups were compared using unpaired two-tailed t-test. **d** The adhesion ability to Caco-2 cells of 3R-ΔvirBR and 3R-ΔvirBR-pUC19-virBR (complemented with intact virBR) compared with 3R parent strain (given as 100%). Data are means ± SEM; n = 3 biologically independent replicates obtained from three independent experiments. One-way ANOVA and Dunnett's multiple comparisons test were performed on values. **e** The adhesion ability to IEC-6 cells of 3R-ΔvirBR and 3R-ΔvirBR-pUC19-virBR (complemented with intact virBR) compared with 3R parent strain (given as 100%). Data are means ± SEM; n = 3 biologically independent replicates obtained from three independent experiments. One-way ANOVA and Dunnett's multiple comparisons test were performed on values. **f.** Transmission electron micrographs illustrating pili formation of 3R, 3R-ΔvirBR, 3R-ΔvirBR-pUC19-virBR. Scale bars, from left to right - 2 μm, 500 nm and 200 nm. Image is representative of three independent experiments. Source data are provided as a Source Data file.

times) of IncX3 plasmid, but not IncX3-ΔvirBR (Fig. 3e) in the presence of 30 mg/L ZnSO₄ or 40 mg/L CuSO₄, as was the expression of virBR (Fig. S16a). Moreover, compared to IncX3-ΔvirBR, both ZnSO₄ and CuSO₄ significantly increased the expression of actX and T4SS genes (virB2 and virD4) in IncX3 (Fig. S16b, c). These results suggest heavy metals activate virBR expression, thereby enhancing the expression of actX and T4SS related genes (such as virB2 and virD4) in IncX3, leading to the increased conjugation frequency of IncX3 plasmid.

To determine the binding site of the actX promoter with VirBR, the upstream region of actX was truncated at -129, -179, -229, and -279 (based on the transcription start site), and fused with gfp to generate pN279-gfp to pN129-gfp. Together with virBR, the fluorescence intensity of pN279-gfp and pN229-gfp was significantly increased but no changes were detected in pN179-gfp and pN129-gfp, suggesting that VirBR is binding to the upstream region of actX between -229 to -179 (Fig. 3b). To determine the specific binding sites of VirBR, we divided the 120 bp upstream region (-279 to -159) of actX into six 20 bp segments. The fluorescence intensity results reveals that VirBR lost function when segment 5 (-199 to -179) was replaced (Fig. 3c, d), indicating that the binding sequence of VirBR is within this 20 bp region of actX, which is AT-rich (AAAAATTTAGT-TAACTTGAC). To further verify that VirBR binds to the actX promoter, electrophoretic mobility shift assays (EMSA) were conducted, and the biotin-labeled actX promoter (used as probe), showed significant lags as the addition of 10-40 μM His6-VirBR, especially when the concentration of His6-VirBR is 20 and 40 μM, whilst the labeled probe was not retarded

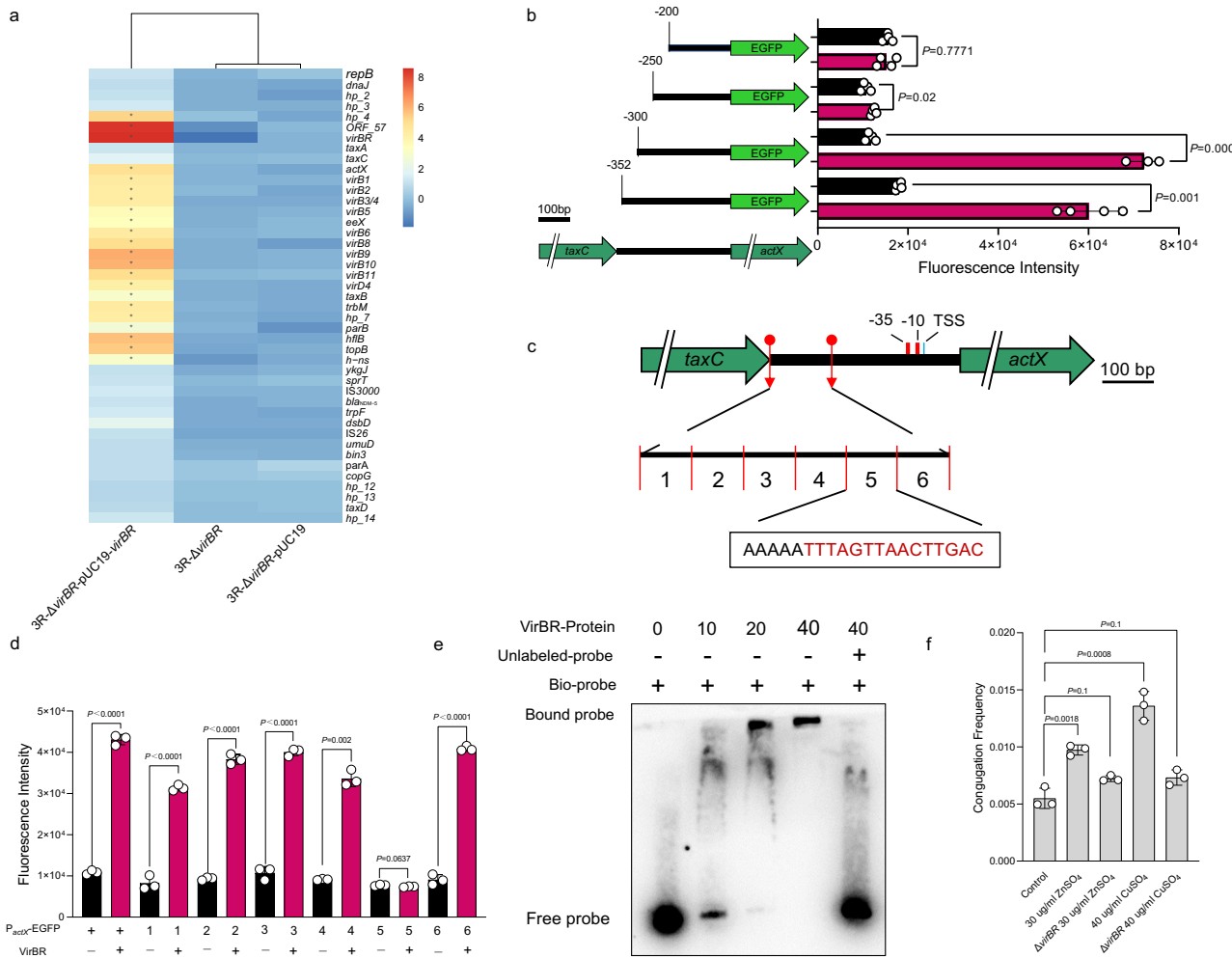

**Fig. 3 | VirBR promotes the transcription of T4SS. a** The fold-change of expression of IncX3 plasmid related genes for 3R-Δ*virBR*-pUC19-*virBR* (complemented *virBR*), 3R-Δ*virBR*, and Δ*virBR*-pUC19 (control). Increase in gene expression appears as yellow-red. **b** Determining the binding region of VirBR immediately upstream of *actX*. Left side, schematic diagram of P*actX* and reporter constructs tagged with EGFP; the mapped promoter is shown in red and TSS in bright blue. Right: Fluorescence intensity of EGFP expression in the absence (black bars) or presence (red bars) of *virBR*. Data are means ± SEM; *n* = 4 biologically independent replicates obtained from three independent experiments. Groups were compared using two-tailed t-test. **c** Schematic diagram of P*actX* regions (1-6 segments) covering between -279 to -159 upstream of the *actX* start codon. **d.** Fluorescence intensity of each of

the segment (described in **c**) with VirBR (red bars) or without VirBR (black bars). Data are means ± SEM; *n* = 3 biologically independent replicates obtained from three independent experiments. Groups were compared using two-tailed t-test. **e** Gel retardation assays of the labelled P*actX*-biotin binding with VirBR. Increasing concentrations of VirBR (0–40 μM) were incubated with biotin label P*actX* fragment. Image is representative of three independent experiments. **f** Conjugation frequency of IncX3 and IncX3-Δ*virBR* in BW25113 in the presence or absence of specific concentration of metal salts. Data are means ± SEM; *n* = 3 biologically independent replicates. One-way ANOVA was performed on values. Source data are provided as a Source Data file.

when co-incubated with 100-fold unlabeled probe (Fig. 3e). These results suggest that VirBR acts as a transcription regulator binding to the promoter region of *actX*, thereby enhancing the transcription of downstream *virB1-11/D4*-related genes. To further confirm the function of *actX*, we constructed and examined the conjugation and colonisation ability of *E. coli* 3R-Δ*actX* and revealed that it not only lost the ability to conjugate in-vitro (Fig. S17a), but also exhibited a significant decrease in colonisation (murine gastrointestinal model) after 11 days post-inoculation which became undetectable at 19 days post-inoculation (Fig. S17b). These results indicated that *actX* plays a significant role in gastrointestinal colonisation of IncX3-carrying bacteria.

## T4SS regulate plasmid conjugation and enhance *E. coli* colonisation

To examine the relationship between the VirB-type T4SS carried by *bla*_{NDM-5}-IncX3 and plasmid conjugation and bacterial colonisation, we knocked out *virB1* (encoding pili synthesis) and *virB2* (encoding related

enzymes of T4SS), which abolished the ability of IncX3 plasmid conjugation transfer, whilst the complementation of *virB1/2* restored its conjugative ability (Fig. S18a). Additionally, the plasmid IncX3-Δ*virB1/2* failed to conjugate to the plasmid-free *E. coli* (Fig. S18b). The cell adhesion ability of *E. coli* 3R-IncX3-Δ*virB1/B2* was also decreased when compared with WT strain (Fig. S18c). Thus, the knockout of *virB1/2* results in the negating T4SS activity. Compared with *E. coli* 3R, the colonisation ability of 3R-IncX3-Δ*virB1/B2* in mice decreased significantly at eight days post-inoculation, and could not be detected at 13 days post colonisation (Fig. S18d). These results indicate that T4SS is associated with both plasmid conjugation and *E. coli* colonisation, and that VirBR can indirectly enhance the transcription of T4SS-related genes by activating the promoter of *actX*.

## Correlation and function of VirBR in IncX family plasmids

*virBR* is 258 bp in length and encodes a protein of 85 amino acids. Its ribosome binding site (RBS) is typical−AGGAGG (Fig. 4a). The protein

a

TAACTTTTGGTTTAATGAGTGTGGTTTAAGTAAATCTGAGG TTATAA ACGAAGTAATTAAC TGGAAGAAT TTCGCATAT
<div align="center">**-35 region**                              **-10 region**</div>
ACCCTTAAAGAACTGGAGGAAGCCAAAAATGAAGCGATAAGAGAACTGCGGAGTTAATCAATTAGGGGTGGTATCAGG
TTGTGGAGGCAACCTGATACCGTGACACTCAATTCCCTCGCGAGGAATATAAAATGTCGGAACGCATGTTATCAGCAA
TACAGACTGTTGAAAAGGGTGGGCGTCCGGTTTTTCCCTTGATGCCATTCTCTGCTTTTCCTGAGTACATGGCATTACT
CAGAAAAGCCCTGGAAAAGAAAGAAACAAAAGCACTGATAGAAAAACAGGAGGTGCT**ATG**AAAAAACAGGAGTATTTT
<div align="center">**RBS**           *virBR*</div>
ATCAGTGCAGAACAGAGGCGGGGAGCTGTCCGGTTTTCGCTGGGATTTAACAGCAAAGGGGAAATTGTATTGCACTG
GACTAACGAGGCCGGGTTAAGAGTCTGGAGTATACTAAGCGGTAACAGGGGAAAAAGTCCCAGCCGGGCAAACCGG
GAAAGAATGAGTAACCTCCGCCGCTGGCTCCATGATGCCCGGCAGGGCATGGAAGGCGACACACCAGAGGCAGAAT
AA

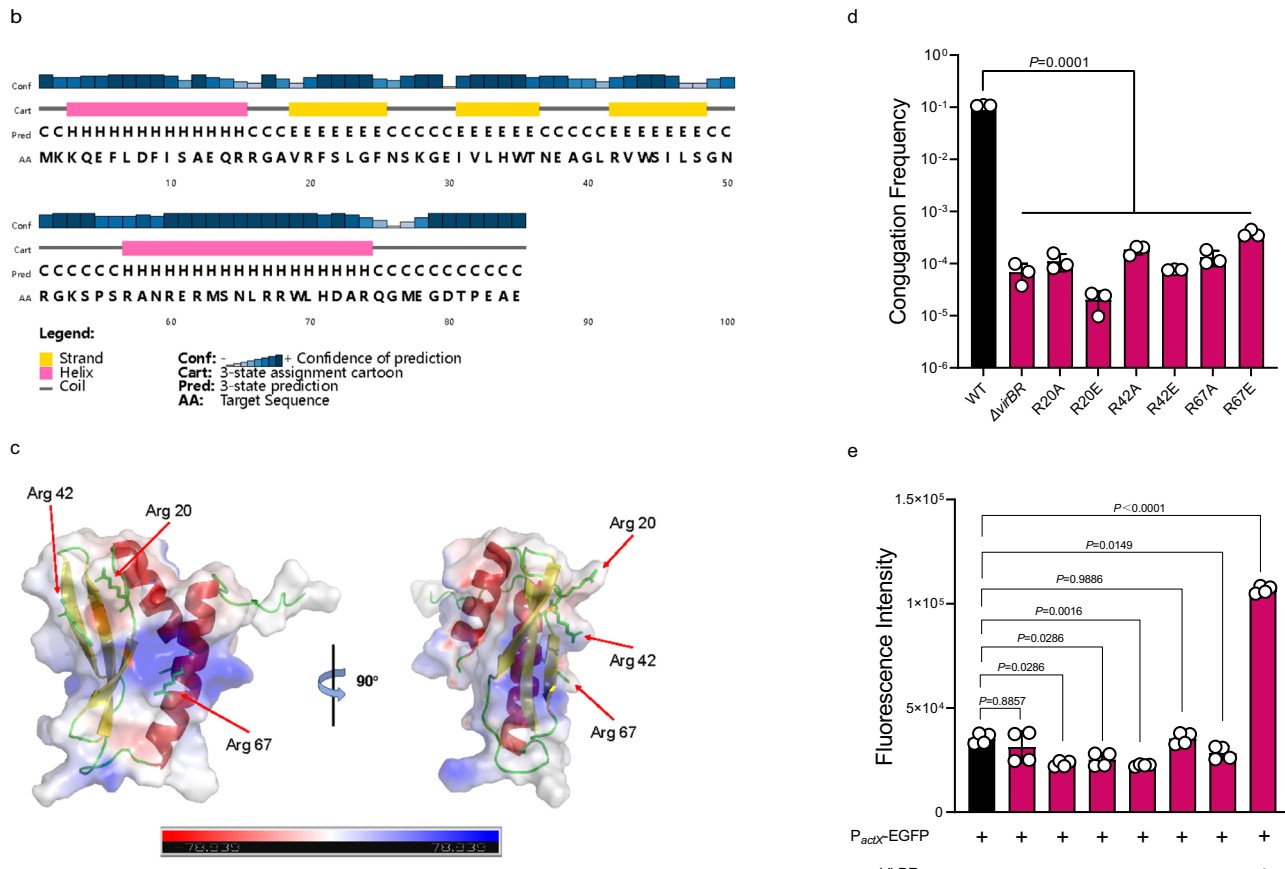

**Fig. 4 | The sequence of *virBR* gene and upstream region, and key amino acids of VirBR. a** Sequence schematic diagram of *virBR* with upstream region. Predict promoter and TSS are shown in red font. Sequence of *virBR* is shown in green font. **b** The predicted secondary structure of VirBR using psipred. The α-helices and β-strands are shown as pink and yellow cylinders, respectively. **c** The predicted protein structure of VirBR using AlphaFold. The key amino acid was shown in Arg 20, Arg 42 and Arg 67. **d** Amino acid substitution analysis on conjugation frequency comparing WT VirBR (black bar) to individual substituted arginine residues (Arg 20, 42 and 67) with either alanine (A) or glutamic acid (E). Data are means ± SEM; $n = 3$ biologically independent replicates. Groups were compared using two-tailed t-test. **e** Fluorescence intensity of $P_{actX}$-EGFP with each of the constructs described in d (red bars) or without VirBR (black bar). Data are means ± SEM; $n = 4$ biologically independent replicates. Groups were compared using two-tailed t-test. Source data are provided as a Source Data file.

structure of VirBR is predicted to be a helix-strand-strand-strand-helix fold using Psipred[21] and AlphaFold2[22], including two α-helixes (α1-2) flanking three β-sheets (β1-3) (Fig. 4b, c). As DNA sequences are negatively charged and can bind to basic amino acids, a total of 26 basic amino acids of VirBR were selected and examined by introducing a series of targeted mutations in pUC19-*virBR* (Fig. 4c). Three key sites were identified - arginine 20, arginine 42 and arginine 67 (R20, R42 and R67), located on β1, β3 and α2, respectively. The conjugation frequency of *bla*~NDM-5~-IncX3 failed to recover when these three amino acids were individually substituted with alanine (R20A, R42A and R67A) or glutamic acid (R20E, R42E and R67E) (Fig. 4d). Subsequently,

these three targeted VirBR substitutions (R20, R42 and R67) were unable to increase the promoter activity of *actX* (Fig. 4e), indicating that R20, R42 and R67 are critical for VirBR activity.

We further examined VirBR homologous in other IncX-families by GenBank database. The VirBR in IncX3 exhibited 60-77.6% amino acid identities with that of other IncX family plasmids (Fig. 5a, b), including IncX1 commonly associated with the mobile colistin resistance gene *mcr-1*, and the tigecycline resistance gene *tet*(X4), IncX2 plasmids often carry the plasmid-mediated quinolone resistance gene *qnr*, and IncX5 plasmids are often associated with carbapenem resistance genes *bla*~KPC-2~ and *bla*~IMP-4~. All VirBR homologues in IncX

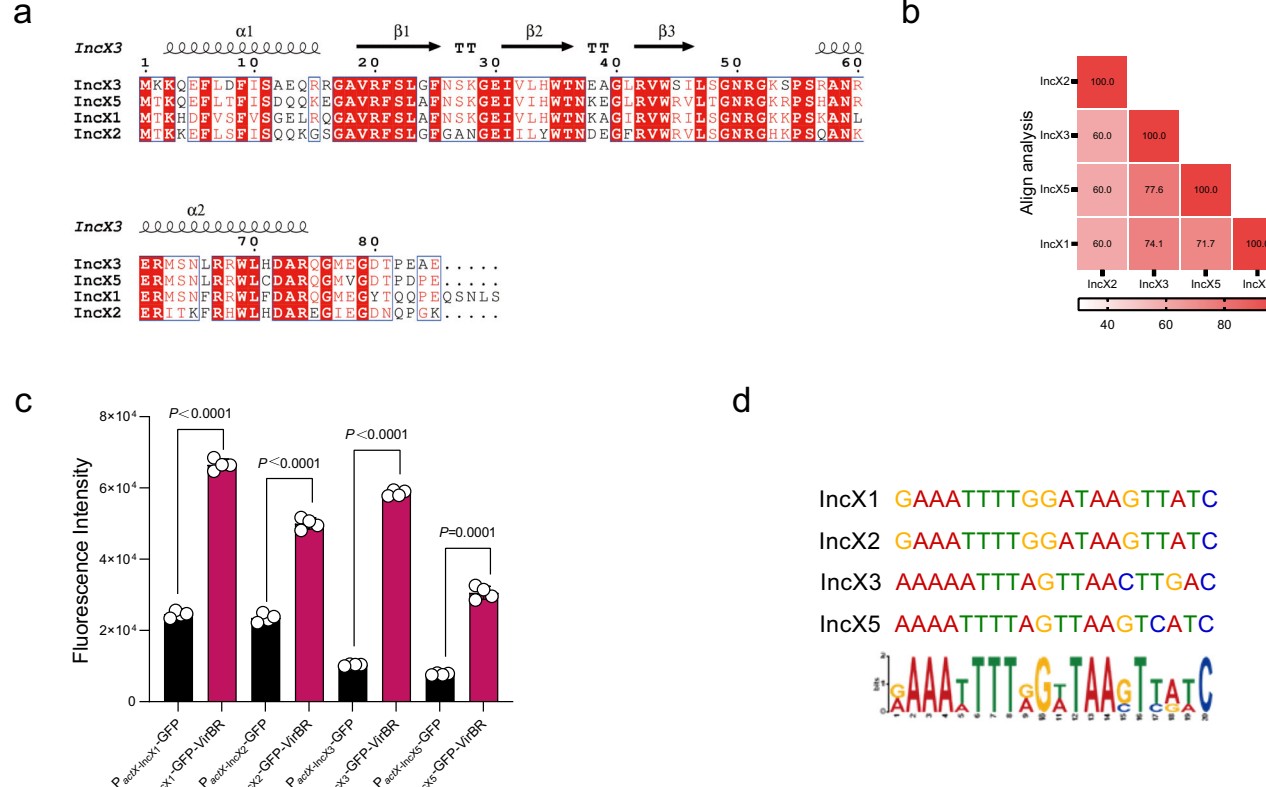

**Fig. 5 | Association of VirBR with IncX family plasmids. a** Multiple-sequence alignment of translated *virBR* genes (VirBR) among various IncX-type plasmids (IncX1 (CP05347.1), IncX2 (LT827129.1), IncX3 (CP049352.1) and IncX5 (KY062156.1)) using CLUSTALW. **b** Sequence alignment percentage using the sequences described in **a**. **c** Function analysis of various VirBR analogues on the expression on GFP labelled *actX*. The fluorescence intensity of P*actX* in IncX1, IncX2, IncX3 and IncX5 plasmid was detected with (red bar) or without (black bar) VirBR. *n* = 4 biologically independent replicates. Groups were compared using two-tailed t-test. **d** P*actX* sequence motif analysis on the VirBR binding region allocated on IncX1, IncX2, IncX3 and IncX5 plasmids. Source data are provided as a Source Data file.

family plasmids possess the same conserved amino acid sequence at the β-sheets (Fig. 5a). To verify the function of these other VirBR homologues, we use the GFP reporter gene to detect the promoter activity of T4SS in IncX1, IncX2 and IncX5. Our results indicate that the VirBR homologues on IncX1, IncX2, IncX3 and IncX5 plasmids can enhance the promoter activity of related *actX*, implying VirBR is a conserved functional transcription regulator in the IncX plasmid family (Fig. 5c). Additionally, a consensus 20 bp VirBR-DNA binding sequence VAAAHTTTVGHTAASTHVHC was also observed in the promoter regions of *actX* on IncX1, IncX2, IncX3, and IncX5 plasmids (Fig. 5d).

## Discussion

Our study examines both exogenous and endogenous factors contributing to the global spreading of carbapenem-resistant *E. coli* carrying *bla*NDM-IncX3. Although carbapenems are prohibited to be used with food-producing animals, over 4000 tons of β-lactams (mainly amoxicillin) and β-lactamase inhibitors are used in animals, which accounts for 12.5% (rank 3rd) of all animal antibiotics used in China mainland in 2020[23]. Additionally, the World Organization for Animal Health reported that β-lactams, mainly penicillins and cephalosporins, account for 15.0% (rank 2nd) of all animal antibiotics used across 109 countries in 2018[24]. These extensively used β-lactams in food-producing animals are likely to select *bla*NDM-5-IncX3-positive *E. coli* persisting in animal gastro-intestinal tracts, as we have demonstrated in this study. Similarly, amoxicillin was proved to be a driver for ESBL colonisation in children[25]. It should be note that amoxicillin is classified by WHO as part of the "Access" list of antibiotics, which are recommended as essential first or second choice treatment options for human infectious syndromes[26]. However, as out study indicates, this classification needs careful reconsideration as amoxicillin has the potential to select and facilitate the spread of carbapenem resistant bacteria—carbapenems are list by the WHO in the "watch" category[26].

Our newly identified transcription regulator, VirBR, binds to the promoter of *actX* gene enhancing T4SS transcription and thus facilitates the IncX3 conjugation and *E. coli* colonisation in the gastro-intestinal tract. Therefore, VirBR, plays a critical role in the intestinal persistence of CRE, as was shown by *bla*NDM-5-IncX3-positive *E. coli* in our in-vivo chicken intestinal models even without antibiotic pressure. Interestingly, Cu and Zn metals play a role in plasmid conjugative transfer by modulating VirBR-associated T4SS, inferring that heavy metals might function as exogenous signals to co-regulate VirBR. It is important to note that Cu and Zn metals are commonly used as trace elements in feed additives to ensure growth, development, and the reproductive performance of livestock and poultry[27,28]. Accordingly, Cu and Zn may act as possible non-antibiotic drivers for the persistence of *bla*NDM-5-IncX3-positive *E. coli* in food-producing animals. CRE's ability to persist in the gastro-intestinal tract is a serious public health concern, not only contaminating the animal production chain and adding further risk to humans through poultry consumption[8], but is also a major threat to immunocompromised humans, including organ and stem cell transplant recipients[29], and endogenous infections via gastro-intestinal tract colonisation[30]. Our study infers that the restricted or prudent use of carbapenems or other β-lactams in clinical

settings has limited impact on CRE-colonised patients to decontaminate CRE; therefore, new strategies are needed for mitigating the impact of IncX3 plasmids carrying $bla_{NDM}$ and other β-lactamase genes in the gastro-intestinal microbiota.

IncX3 plasmid-borne VirBR, promoting the transcription of T4SS and thus enhancing persistence extends our understanding on the function of T4SS. The IncX3 plasmid-derived T4SS belongs to *Agrobacterium tumefaciens* VirB/D4 type that generally composes of 12 core genes *virB1–virB11/virD4*[31] and serves as a paradigm for type IVA systems[32]. These IVA systems played a vital role in the contact-dependent secretion of effector molecules into eukaryotic hosts, transfer of mobile DNA elements, and contact-independent exchange of DNA with the extracellular milieu[33]. Although T4SS was reported to be related with biofilm formation to promote the gastro-intestinal persistence of adherent-invasive *E. coli*[34], the regulation of T4SS and its role on the global spread of $bla_{NDM-5}$-IncX3 plasmid hasn't been fully demonstrated. Herein, we confirm the function of VirBR by regulating the transcription of T4SS, as well as the role of T4SS in enhancing *E. coli* colonisation. VirBR-mediated T4SS regulation is associated with the gastro-intestinal persistence by CRE strains through two possible ways. First, the conjugation ability of IncX3 plasmid as the cell-envelope-spanning T4SS machine facilitating the mating channel and extracellular pili is critical for conjugation[33]. Transcription regulator VirBR promotes the transcription of T4SS to enhance conjugation transfer levels of IncX3 plasmids which can also enhance the plasmid spreading in bacterial populations. Second, VirB2 and VirB5 constitute the outer membrane portion of pili, with VirB5 being recruited to VirB2 to form the pili tip, enabling it to bind to specific cell surface receptors or directly embed into the target cell membrane[35]. Here we found that VirBR plays up-regulating role in maintaining T4SS pili formation, as the TEM confirmed that the T4SS pili of *E. coli* 3R were reduced in Δ*virBR* and Δ*virB1/2* backgrounds, and restored via complementary intact genes (Fig. 2f), indicating that the pili mediated by VirBR can adhere to the cell, contributing to the persistence of bacterial colonisation. Similarly, another type of T4SS, *tra*-T4SS was proved to play a crucial role in persisting the colonisation of adherent-invasive *E. coli* in the gut, which suggests that the T4SS can increase the biofilm formation to promoting bacteria the colonisation ability[34].

VirBR is identified as a transcriptional regulator experimentally that binds the promoter of *actX* to enhance the transcription of T4SS. However, VirBR does not possess typical DNA-binding domains such as ribbon-helix-helix (RHH) or helix-turn-helix (HTH), which binds the double stranded DNA molecule in the major groove[36]. Notwithstanding, the possible function region of VirBR has been verified by key amino acid substitutions and mechanistic studies, that is, the positive-charged surface containing β1, β3 and α2 is essential for the function of VirBR. These observations enhance our understanding of transcription regulators and their role in the global prevalence of IncX3 plasmids. Nevertheless, the detailed mechanistic analysis of this atypical transcription regulator, VirBR, binding to DNA necessitates further studies. Homologue analysis reveals that VirBR has high amino acid sequence identity (60.0-77.6%) to that of other IncX family plasmids, including IncX1 plasmids carrying *mcr-1*[37], *tet*(X4)[38], and/or ESBL[39] genes; IncX2 plasmids are associated with *qnr*, and IncX5 plasmids with $bla_{KPC-2}$ and $bla_{IMP-4}$. Thus, VirBR-like regulators are important regulatory elements of T4SS and facilitate the persistence of bacteria carrying these extensively-drug resistant (XDR) plasmids. The IncX family plasmids were considered as 'narrow' host-range but our data suggests their host range might be greater than previously appreciated[40], further highlighting the importance of IncX-type plasmids in the dissemination of XDR genes.

In summary, we observed that β-lactams amoxicillin can promote $bla_{NDM}$-IncX3-positive *E. coli* persistence in chicken gastro-intestinal tracts which indicated merely prohibiting the use of carbapenems in livestock farming is insufficient to combat carbapenem-resistant

bacteria. Moreover, we have identified a transcription regulator gene, *virBR*, on IncX3 plasmids that can enhance T4SS transcription and promote plasmid conjugation and bacterial colonisation. The universal presence and high homology of VirBR-like proteins in IncX family plasmids facilitating the spread of XDR genes indicating their importance and potential global impact on both human and animal health.

## Methods

### Ethical approval statement
The chickens and mice were raised and handled in compliance with the Chinese laws and guidelines (protocol GKFCZ2001545), EU Directive 2010/63/EU for animal experiments, and the China Agricultural University regulations concerning the protection of animals used for scientific purposes (2010-SYXK-0037). Committee on Animal Welfare and Ethics in China Agricultural University approved the chickens and mice experiments (AW32803202-2-1 and AW32803202-2-2).

### Chickens
All experiments were carried out using 1-day-old ROSS chickens purchased from a commercial hatchery in Hebei province. The chickens were housed in sterile isolators at the Laboratory Animal Public Service Platform of China Agricultural University with antibiotic-free feed and water, which had been detected by UPLC-MS-MS[41]. Before breeding and inoculation, ten chickens were euthanized to collect intestinal contents, and all chickens had cloacal swabs collected to culture onto brain heart infusion agar (BHA) plates (containing 0.25 mg/L meropenem). DNA was extracted from all chicken samples to perform PCR and RT-qPCR[42] to confirm that they were $bla_{NDM}$-negative. Simultaneously, the chicken feed, drinking water, and isolators were subjected to the same testing procedure to ensure that they were also $bla_{NDM}$-negative. All chicken handling, breeding, and intestinal tract collection were performed in biosafety level 2 facility while wearing protective clothing, surgical gowns, sterile gloves, and masks.

### Mice
Female BALB/c mice, aged 6–8 weeks, were purchased from SPF (Beijing, China) Biotechnology Co., Ltd. Mice were raised in a specific pathogen-free room for 1 week to prevent a stress reaction, with antibiotic-free feed and water. The light cycle was set to 12 h of light and 12 h of darkness, with a temperature range of 18–23 °C and humidity maintained between 40% and 60%. For three days prior to colonization, mice were orally gavaged with 100 μL of a mixture containing ampicillin (2500 mg/L), streptomycin (500 mg/L), vancomycin (500 mg/L), metronidazole (2500 mg/L), and kanamycin (400 mg/L) to deplete the indigenous gut microbiota.

### Bacterial strains, plasmid, and growth conditions
The bacterial strains and plasmid used in this study are listed in Supplementary Table 3. The *E. coli* 3R and *E. coli* BW25113 strains were grown in Luria-Bertani (LB) broth (Sigma-Aldrich) at 37 °C with shaking (200 rpm) or on LB agar plates. Antibiotics were added at the following concentrations for both conjugation experiments and genetic tool editing in the experiments: 50 mg/L kanamycin, 2 mg/L meropenem, 50 mg/L apramycin, 50 mg/L spectinomycin, 50 mg/L chloromycetin, and 50 mg/L sodium azide. All strains were stored in 20% glycerol at −80 °C.

### Inoculated strains
The inoculation strains, *E. coli* 3R (678) and *E. coli* 8R (622), were isolated from chickens' cloacal swabs, as reported in our previous study[8]. The minimum inhibitory concentrations (MICs) were performed on these two inoculation strains using the agar dilution method, following the guidelines provided in the Clinical and Laboratory Standards Institute document (CLSI, 2017) and the European Committee on Antimicrobial Susceptibility Testing (EUCAST, version 6.0). As a quality control, the reference strain *E. coli* ATCC25922 was used.

## Experimental design

Figure 1a: a total of 74 1-day-old chickens were randomly divided into 2 groups, one is 3R group, another is 3R+amoxicillin group. From 6-old-day to 40-old-day, chickens of 3R+amoxicillin group were treated with 1 mg/L amoxicillin in water. Each group was oral gavage with 1 mL 10$^8$ CFU/mL of 3R at chicken 8-day-old. Four chickens of each group were euthanized and cecum were taken to quantify the inoculated strains at chicken 10, 11, 13, 15, 19, 26, 33, 40, 47-day-old. Four chickens of each group were euthanized and cecum were collected at chicken 10, 13, 19, 26, 33, 40, 47-day-old. Then do the quantification of NDM-positive strains and non-inoculated NDM-positive strains (Please refer to the following for detailed steps). Figure 1b: a total of 106 1-day-old chickens were randomly divided into 3 groups, one is 3R group, one is 8R group, and the other is control group. Chickens of each group were oral gavage with 1 mL PBS, 10$^8$ CFU/mL of 3R or 8R strain at 8-day-old. Four chickens of each group were euthanized and duodenum, jejunum, ileum, cecum, and rectum were collected at chicken 10, 11, 13, 15, 19, 26, 33, 40, 47-day-old. Then do the quantification of NDM-positive strains and non-inoculated NDM-positive strains. Then do the quantification of NDM-positive strains, non-inoculated NDM-positive strains, $bla_{NDM}$ gene and whole genome sequencing of strains (Please refer to the following for detailed steps). Fig. S10a: a total of 30 1-day-old chickens were randomly divided into 2 groups, one is 3R group, another is 3R-Δ$virBR$ group. The chickens of each group were oral gavage with 1 mL 10$^8$ CFU/mL of 3R or 3R-Δ$virBR$ at 8-day-old. Three chickens of each group were euthanized and cecum were taken to quantify the inoculated strains at chicken 10, 11, 13, 15, 19-day-old. Fig. S10b: a total of 12 7-week-old mice were randomly divided into 2 groups, one is 3R group, another is 3R-Δ$virBR$ group. The mice of each group were oral gavage with 0.1 mL 10$^8$CFU/mL of 3R or 3R-Δ$virBR$, then collected mice feces to quantify the inoculated strains every day.

## Inoculation proposal

The two inoculation strains were grown at 37 °C in Brain Heart Infusion until reaching the stationary phase. They were then diluted in phosphate-buffered solution (PBS) to a concentration of 10$^8$ colony-forming units (CFU). The chickens of colonisation groups were orally administered 1 mL of the diluted solution, which contained $1.0 \times 10^8$ CFU of either *E. coli* 3R or *E. coli* 8R, using oral gavage[43].

## Amoxicillin treatment proposal

From 6-old-day to 40-old-day, the chickens in the *E. coli* 3R with amoxicillin group were treated with 1 mg/L of amoxicillin in their drinking water.

## Quantification of NDM-positive strains

The total weight of the contents of the five intestines was measured individually, and then 0.2 g from each intestine was resuspended in 900 μL of PBS. The suspensions were diluted in a 10-fold gradient and spread onto the Eosin Metylene Blue (EMB) agar plates supplemented with 30 mg/L vancomycin and 0.25 mg/L meropenem. The plates were incubated overnight at 37 °C. EMB agar is specifically used to isolate gram-negative enteric bacteria, especially coliform and fecal coliform group. Using the standard colony counting method, the colonies grown on the agar plates were counted to determine the total number of meropenem-resistant colonies in each intestinal content. Subsequently, on the EMB agar, 50 colonies were randomly selected from each intestine sample and streaked onto LB agar plates. The plates were then incubated overnight at 37 °C. For each colony grown on LB agar plates, the presence of the $bla_{NDM}$ gene was determined by PCR to obtain the ratio of NDM-positive colonies to meropenem-resistant colonies. Lastly, the total number of NDM-positive colonies in each intestine sample was calculated.

## Quantification of non-inoculated NDM-positive strains

To distinguish between inoculated strains (3R and 8R) and transfer strains (non-inoculated strains acquiring $bla_{NDM}$ from 3R or 8R), the presence of the $bla_{CTX-M-65}$ resistance gene was verified by PCR. Since the $bla_{CTX-M-65}$ gene is exclusively present on the chromosomes of 3R and 8R but not on the plasmids, it serves as a marker to differentiate between inoculated strains and transfer strains.

## DNA extraction and quantification of $bla_{NDM}$ gene

Approximately 0.2 g of cecal and rectal contents from broilers at 1, 8, 10, 13, 19, 26, 33, 40, and 47 days old were separately collected for genomic DNA extraction using the DNeasy PowerSoil Kit (QIAGEN, Germany). The abundance of $bla_{NDM}$ gene in the genomic DNA of each sample at each time point was quantified by SYBR Green-based real-time quantitative PCR. The real-time PCR assay was performed on the QuantStudioTM 7 Flex Real-Time PCR System (Applied Biosystems), and the quantification was normalized using 16 S rRNA as an internal control[44]. The relative copy numbers of $bla_{NDM}$ gene in the cecal and rectal contents of broilers at different ages were determined.

## Plasmids and strains construction

The bacterial strains, plasmid and primers used in this study are listed in Supplementary Table 3 and Supplementary Table 4. Deletion mutants of p3R-4 plasmid were generated by the λRed recombination system[45]. To facilitate manipulation, the resistance gene of pUC19, pKD46 and pCP20 were replaced by *aacC4* or *aadA*. The replace template, amplified using $virBR$-del-F/R primers with pKD3, was electroporated to *E. coli* 3R competent cell carrying the p3R-4 plasmid to replace the $virBR$ gene with *cat*. The deletion mutants were confirmed by sequencing using $virBR$-F/R primers. The wild-type plasmid p3R-4 and its derivative plasmid p3R-4 Δ$virBR$ were then electroporated into *E. coli* BW25113. The $virBR$ gene with its native promoter was amplified from p3R-4 using primers pUC-$virBR$-F/R and pAC-$virBR$-F/R, respectively. Additionally, the complementary vector of pUC19 and pACYC184 were amplified from pUC19 and pACYC184 plasmids using primers pUC19-v-F/R and pACYC184-v-F/R, respectively. Then pUC19-$virBR$ and pACYC184-$virBR$ plasmids were generated by cloning purified $virBR$ PCR products into pUC19 and pACYC184, respectively. Similarly, using the aforementioned λRed recombination system, we replaced the $bla_{CTX-M}$-65 gene on the 3R chromosome with the *cat* gene. The fragment containing the upstream region of *actX* was amplified from p3R-4 using *actX*-p-F/R primers and cloned into the pGFP plasmid, which contains a promoterless *gfp* gene, to produce p*actX*-*gfp* plasmid. Base on p*actX*-*gfp*, $virBR$ was amplified from p3R-4 using $virBR$-g-F/R primers and cloned into p*actX*-*gfp*, generating the p*actX*-*gfp*-$virBR$ plasmid. Using p*actX*-*gfp* and p*actX*-*gfp*-$virBR$ as templates, plasmids p*taxA*-*gfp*, p*virB1*-*gfp*, p*taxA*-*gfp*-$virBR$, and p*virB1*-*gfp*-$virBR$ were constructed by employing *taxA*-g-F/R and *virB1*-g-F/R primers, following the aforementioned methods. The ORF of $virBR$ was amplified from p3R-4 using primers $virBR$-protein-F/R and cloned into pET28a to generate the expression vector pET28a-His-$virBR$.

## RNA-seq

Overnight cultures of 3R and its derivatives were diluted 1:100 in 20 mL of LB broth without antimicrobials. After incubation at 37 °C and 200 rpm for 5 h, the cultures were harvested and resuspended in PBS (0.01 M, pH7.4). RNA-Seq and RNA-Seq data analysis were conducted following previously reported methods[46].

## Competition experiments in-vitro

In-vitro competition experiments were conducted to evaluate the relative fitness of 3R and 3RΔ$virBR$, as previously described[18]. Briefly, 3R and 3RΔ-$virBR$::*cat* were cultured overnight and adjusted to a 0.5 McFarland standard in LB broth. Then the 1:100 dilutions were mixed

at 1:1 ratio, and the mixtures were cultured for 24 h. Subsequently, 10 μLof the mixture was further mixed in 990 μL of LB broth and continued to be cultured. The mixtures at 24, 72, and 120 h were diluted and plated on LB agar containing meropenem (0.5 mg/L) or chloramphenicol (50 mg/L) to count the colony-forming units (CFU). The relative fitness was calculated by dividing the CFU of 3R-*virBR*::*cat* by the CFU of 3R.

### Plasmid invasion assays

Plasmid invasion assays were used to evaluate the competition favors of p3R-4 and its derivatives according to the previous study[47]. Briefly, BW25113-IncX3 (BW25113-IncX3Δ*virBR* or BW25113-IncX3Δ*virB1/2*) were adjusted to 0.5 McFarland standard, then diluted 1:100 into 1 mL LB broth. The BW25113-IncX3 (BW25113-IncX3Δ*virBR* or BW25113-IncX3Δ*virB1/2*) diluent were mixed with BW25113 overnight cultures at 1:1 ratio, 100 μL of the mixtures were dripped on filters placed on LB agar. The co-cultures were collected and diluted 1:100 in to 1 mL LB broth for passage culturing per 24 h. The mixtures at 0, 24, 48, 72, 96, 120 h were diluted and plated on LB agar containing meropenem (0.5 mg/L) to count the CFU of BW25113-IncX3 (BW25113-IncX3Δ*virBR* or BW25113-IncX3Δ*virB1/2*), meanwhile, the diluents were plated on antimicrobial-free LB agar to count the total CFU. For the co-culture experiments of BW25113, BW25113-IncX3, and BW25113-IncX3-Δ*virBR*::*cat*, the dilution strategies were similar as above, but the mixtures were composed of 25 μL-BW25113-IncX3, 25 μL-BW25113-IncX3Δ*virBR*::*cat*, and BW25113 diluent. The co-cultures at different dilution gradient were plated on LB agar containing meropenem (0.5 mg/L) or chloramphenicol (50 mg/L) to count the CFU of BW25113-IncX3 or BW25113-IncX3-Δ*virBR*::*cat*. Total bacteria amounts were counted by plating the co-cultures on antimicrobial-free LB agar.

### Plasmid conjugation assays

Plasmid conjugation assays were performed using *E. coli* J53 as the recipient strain, and *E. coli* 3R and BW25113-IncX3 or its derivatives as the donor strains. For solid mating, bacteria cells at the logarithmic phase (OD600 = 0.5-0.6) were mixed at a donor-to-recipient ratio of 1:3 (v/v). Then, a 50 μL mixture was spotted on a nitrocellulose membrane on LB agar and incubated at 37 °C for 12 h. For liquid mating, the cells were mixed at a donor-to-recipient ratio of 1:1 (v/v), and 1 mL mixture was incubated at 37 °C for 12 h. After incubation, the total colony-forming unit (CFU) of recipients and transconjugants were counted on LB agars containing 100 mg/L sodium azide or 100 mg/L sodium azide with 0.5 mg/L meropenem, respectively. Plasmid conjugation transfer frequencies were calculated as the number of transconjugants per donor.

### Plasmid conjugation assays under antibiotic pressure

According to the previous study[48,49], inoculate overnight cultures of donor strains 3R, 3RΔ*bla*CTX-M-65, and recipient strain J53 into 1 mL of antibiotic-free LB broth. Incubate at 37 °C with shaking at 200 rpm for 5–6 h. The MIC values for donor strains 3 R and 3RΔ*bla*CTX-M-65 against meropenem and amoxicillin are 128 mg/L and 6400 mg/L, respectively. Incubate donor strains with 1/2 MIC drug concentrations for 6 h, followed by three washes with PBS to remove the drugs from the bacterial suspension. Subsequently, mix 20 μL and 60 μL of donor and recipient bacterial suspensions, respectively, at a volume ratio of 1:3. Take 50 μL of the mixed bacterial suspension and spot it onto sterile filter membranes placed on antibiotic-free LB agar plates. Incubate overnight at 37 °C. The next day, perform a gradient dilution by washing the bacterial cells from the filter membrane into 900 μL PBS. Apply appropriate dilutions to LB agar plates containing 2 mg/L meropenem and LB agar plates containing 2 mg/L meropenem plus 50 mg/L sodium azide. Incubate for 16–24 h. Finally, record the bacterial colony counts on the plates and calculate the plasmid conjugation transfer efficiency using the formula (conjugants/recipient bacteria) × 100%.

### EMSA−Electrophoretic mobility shift assay (EMSA)

EMSA assays were performed using the LightShift™ Chemiluminescent EMSA Kit (Pierce, USA) according to the manufacturer's instructions. The biotin labeled $P_{actX}$ DNA was design and synthesized following the map of IncX3 plasmid. DNA binding reaction was performed in 20 μL system, which consisted of $P_{actX}$ probe, VirBR protein, 1×binding buffer, 2.5% glycerol, 5 mM MgCl$_2$, 50 ng/μL poly(dI•dC), and 0.05% NP-40. The binding mixtures were incubated in room temperature for 20 min then loaded onto a 6% polyacrylamide gel in 0.5×TBE. The mixtures were then subjected to electrophoresis in 4 °C under 110 V for 1 h. Biotin-labeled probe was transferred onto a positively charged nylon membrane and underwent UV-crosslink. Streptavidin-HRP was used to label the probe-protein complex and chemiluminescence was adopted to detect the probes.

### GFP reporter gene assays

DH5α bacteria carrying *gfp* fusion vectors were cultured to an OD600 = 0.5, then harvested and resuspended in PBS to an OD600 = 0.5. The fluorescence of cells was quantified using a multimode plate reader (TECAN) with excitation and emission wavelengths of 488 and 525 nm, respectively. The fluorescence intensity was calculated by formula: promoter activity=fluorescence intensity/OD600[50].

### Cell adhesion assays

Adhesion assays for stains were performed using human colon adenocarcinoma cells Caco-2 cells from Kunming Cell Bank of Chinese Academy of Sciences (KCB200710YJ) and Rat small intestine cell IEC-6 from Shanghai Fuxiang Biotechnology Co., LTD (ATCC CRL-1592) according to standard methods[51]. Briefly, Caco-2 and IEC-6 cell were infected with different strains at a multiplicity of infection (MOI) of 20:1. After 3 h, cells were washed with sterile 1×PBS three times to remove non-adherent bacteria. Subsequently, the cells were lysed using 1% Triton X-100 in PBS, and the lysates were serially diluted and plated on selective media. The results were obtained from three biological replicates.

### Transmission electron microscopy (TEM)

The TEM were performed using the Electron Microscopy (EM) Facility at the Dunn School of University of Oxford. Firstly, bacterial strains (3R, 3R-Δ*virBR*, 3R-Δ*virBR*-*virBR*) were cultured on LB agar plate for 12 h, and carbon-coated 300 mesh Cu grids were first subjected to glow discharge and then positioned with the carbon side facing down onto a cluster of colonies that had grown overnight on an agar plate. Gentle pressure was applied to the edges of the grid to ensure the adhesion of bacteria to the carbon film. Subsequently, the grid was transferred (with the carbon side facing down) onto a 25 uL droplet of 4% PFA in PBS and incubated at room temperature for 15 m. Afterward, the grid was washed by briefly passing it over 3 × 25 uL droplets of PBS. Following the wash, the grid was stained with 2% uranyl acetate for 10 s, briefly blotted, and then left to air dry. Finally, the grids were imaged using a JEOL 1400 TEM operated at 120 kV, equipped with a Gatan Rio camera.

### Protein expression and purification

BL21 bacteria carrying the pET28a-His-*virBR* plasmid was grown in LB broth at 37 °C until reaching an optical density at 600 nm (OD600) of 0.3. Induction of protein expression was achieved by adding 1 mM isopropyl-β-D-thiogalactopyranoside (IPTG) to the culture, and the cells were incubated for 12 h. Cells were harvested and resuspended in lysis buffer containing 50 mM NaH$_2$PO$_4$, 500 mM NaCl, 5 mM imidazole (pH 8.0). The cells were lysed by sonication on ice, and the lysate was then centrifuged at 12,000 g for 30 min. The supernatant containing the protein of interest was filtered and collected. The protein solution was incubated with Ni-NTA resin (from GE, USA) following the

manufacturer's instructions, allowing the His-tagged VirBR protein to bind to the resin. After binding, the resin was washed to remove unbound contaminants. Elution of the purified protein was performed, typically using an imidazole gradient. Protein concentrations were detected using BCA assay kit from Thermo Fisher Scientific, USA.

## Bioinformation analysis

Genome and plasmid sequences were obtained from NCBI GenBank database. IncX plasmids multiple sequence alignment was done using BLASTn[52] and visualized using EasyFig version 2.2.2[53]. VirBR protein homologs alignments were done using CLUSTALW[54] and visualized using ESPript 3.0[55].

## Statistics and reproducibility

Statistical analyses were performed for each experiment, and specific details were provided in the figure legends. GraphPad Prism software (version 9.3.1) or R (version 4.1.2) was used for statistical analysis. The $P$ values represented in each figure.

## Reporting summary

Further information on research design is available in the Nature Portfolio Reporting Summary linked to this article.

## Data availability

The genome sequencing data of 3R were deposited in GenBank and are registered under Accession No. SAMN14134968. The genome sequencing data of 8R were deposited in GenBank and are registered under Accession No. CP110410-CP110414 (https://www.ncbi.nlm.nih.gov/nuccore/CP110410.1; https://www.ncbi.nlm.nih.gov/nuccore/CP110411.1; https://www.ncbi.nlm.nih.gov/nuccore/CP110412.1; https://www.ncbi.nlm.nih.gov/nuccore/CP110413.1; https://www.ncbi.nlm.nih.gov/nuccore/CP110414.1). The RAN-sequencing raw data reported in this paper have been deposited in the Genome Sequence Archive (Genomics, Proteomics & Bioinformatics 2021) in National Genomics Data Center (Nucleic Acids Res 2022), China National Center for Bioinformation / Beijing Institute of Genomics, Chinese Academy of Sciences (GSA: CRA015156) that are publicly accessible at. Source data are provided with this paper.

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

## Acknowledgements

This work was supported in part by grants from the National Natural Science Foundation of China (32141002 to S.J.Z., 81991535 to W.C.M., and 32202868 to M.T.F.), the Guangdong Major Project of Basic and Applied Basic Research (2020B0301030007 to W.Y.), and the UK Medical Research Council (project DETER-XDR-China-HUB, grant number MR/S013768/1 to T.R.W.). The funders had no role in study design, data collection, and interpretation, or the decision to submit the work for publication.

## Author contributions

T.M. and N.X. contributed equally to this work. Y.W., T.R.W., and J.S. designed the study. T.M., N.X., Y.G., J.F., and Y.W. wrote the manuscript. Y.W., T.R.W., J.S., and J.P. revised and perfected the manuscript. T.M., J.F., and N.X. constructed the chicken colonisation model. N.X., Y.G., T.M., and C.T. constructed the mice colonisation model and performed investigation of VirBR mechanism and Q.J. and Q.Y. guided the knocked-out experiments. C.W., S.W., and Z.S. guided the chicken colonisation model. All authors read and approved the manuscript.

## Competing interests

The authors declare no conflict of interest.
