## [Peer Review File · Nature Communications]

REVIEWER COMMENTS

Reviewer #3 (Remarks to the Author):

Concerning the answers to my own comments, they are satisfactory but one. I suggested that they compared in vivo the enhancing effect of a carbapenem on the transfer rate of their target gene which confer resistance to carbapenem (which they did not do) to that of Amoxicillin (which they did). I thought that it would be very interesting to know if the enhancing capacity of the antibiotic for which the gene encodes resistance (carbapenem) is greater or lower in comparison to that of the much older and broadly used antibiotic Amoxicillin. They answered that my request was somewhat irrelevant because they are interested in what happens in animals where carbapenem are not used. I agree on that but they experiment could be of great interest for better understanding the dissemination of the target gene in human populations (in whom carbapenems are used) and in which the gene and its regulatory system will very certainly disseminate at some point from animals to humans. I admit that however interesting this experiment might be for the general knowledge and understanding of the mechanisms of dissemination of carbapenem resistance globally, it may be considered a bit borderline for the scope of the paper. However the authors also answered my comment by saying that they had in vitro experiments that could answer the question but they did not show the results which seems odd to me.... “ See below”. Therefore they show these results in the new publication in comparison with those of amoxicillin.

Reviewer #4 (Remarks to the Author):

Reviewers' comments (remarks to the author)

In the reply to the Editor's comments authors included four points summarizing the aspects of novelty in their study. And these aspects are also included and discussed in the revised version of the manuscript.

- "The veterinary used β -lactam antibiotic amoxicillin can enhance the conjugation and persistence of bla_{NDM-5}-IncX3-carrying E. coli in the chicken gut".

Amoxicillin does not enhance conjugation. The word "enhance" should be changed with "select".

The animal model formally demonstrates that amoxicillin is sufficient to maintain the IncX3-blaNDM-5 population, and there is no need to use carbapenems. This result is expected since the blaNDM-5 has a large spectrum against beta-lactams. Therefore, amoxicillin can sustain the colonization of the NDM-5 producers. Furthermore, the presence of the blaCTX-M-65 resistance gene on the chromosomes of 3R and 8R strains also plays its role in sustaining the growth of these strains in amoxicillin-treated animals. In this context, the discussion and conclusions about the conjugation and stability of the IncX3 vehicle obtained in the chicken gut model comparing amoxicillin-treated animals with untreated animals should consider the positive selection on amoxicillin-resistant strains.

It is not clear if experiments in panels E and F of Figure 1 were performed in presence or absence of amoxicillin.

- "IncX3-carrying *E. coli* can persist in the chicken gut in the absence of antibiotic, and the novel IncX3 plasmid-borne virBR gene was identified to be responsible for the gut persistence of blaNDM-5-IncX3-carrying *E. coli*".

The work adds novel information on identification and regulation of the T4SS of the IncX3 plasmids. Previous studies on adherent-invasive *E. coli* demonstrated that T4SS plays a role in the gut colonization [29]. The role of pili production produced under VirBR regulation should be more clearly stated, otherwise it seems that the entire conjugative process is linked to persistence of the *E. coli*.

- "The heavy metal feed-additives, such as Cu and Zn ions play a key role as an exogenous signal in plasmid conjugative transfer by regulating VirBR-associated T4SS, therefore promoting the persistence of IncX3 plasmid-carrying *E. coli* in the absence of antibiotic. This dataset is in addition to the previous submission".

These conclusions are based on *in vitro* conjugation efficiency and mRNA measurements. There is no evidence that these metals interfere with the transcription. At least there is not sufficient strong demonstration to include this aspect in the title of the article. The Cu and Zn effect was not tested *in vivo*. I strongly suggest including this evidence in the text but not in the title, it is not the major outcome of the study. It is overstated.

Reviewer#1 comments

1- "We will explore in detail how heavy metal ions regulate virBR gene expression in further studies as our current work mainly focused on the identification, characterization and function of VirBR". The heavy metal ions regulation has not been explored. In the revised version the activation by those exogenous signals is presented as mRNA abundance of VirBR and T4SS in the presence or absence of ZnSO₄ or CuSO₄. This is a preliminary result despite it is an interesting observation.

2. The response of the authors is acceptable. They provided several additional experiments including actX promoter experiments and His6-VirBR assays and better EMSA experiments. However, the experiment with BW25113 demonstrates that it should be added conjugation efficiency could be “donor strain dependent”.

The response to the third, fourth and fifth major concerns are acceptable.

Could VirBR promote persistence not through the demonstrated effects on tra gene expression but by regulating another plasmid- or chromosomally encoded fitness trait?

Response: Please see our response to your third major comment listed above.

The pili role in adhesion should be better specified.

Reviewer#2 comments

- The reply “the veterinary used β -lactam antibiotic amoxicillin can enhance the conjugation and persistence of blaNDM-5-IncX3-carrying E. coli in the chicken gut” is not acceptable. The most important problem here is that this sentence suggests that the authors are describing the amoxicillin as a positive regulator of the conjugative cluster, and as inducer of persistence. The E. coli cells carrying the plasmid in these experiments are resistant to amoxicillin, while the endogenous gut flora is not. Therefore, an higher levels (pink boxes) in Fig 1.b) are expected by the positive selection of the 3R strain. Which is the experiment demonstrating that amoxicillin enhances in vitro the conjugation efficiency of the 3R donor?

-The reply to the question “Which is the other b-lactam resistance genes associated with NDM-5 gene on the same Inc plasmid?” should be extended with a consideration that the 3R and 8R strains carried a blaCTX-M gene in the chromosome.

REVIEWER COMMENTS

Reviewer #3 (Remarks to the Author):

Concerning the answers to my own comments, they are satisfactory but one. I suggested that they compared in vivo the enhancing effect of a carbapenem on the transfer rate of their target gene which confer resistance to carbapenem (which they did not do) to that of Amoxicillin (which they did). I thought that it would be very interesting to know if the enhancing capacity of the antibiotic for which the gene encodes resistance (carbapenem) is greater or lower in comparison to that of the much older and broadly used antibiotic Amoxicillin. They answered that my request was somewhat irrelevant because they are interested in what happens in animals where carbapenem are not used. I agree on that but they experiment could be of great interest for better understanding the dissemination of the target gene in human populations (in whom carbapenems are used) and in which the gene and its regulatory system will very certainly disseminate at some point from animals to humans. I admit that however interesting this experiment might be for the general knowledge and understanding of the mechanisms of dissemination of carbapenem resistance globally, it may be considered a bit borderline for the scope of the paper. However, the authors also answered my comment by saying that they had in vitro experiments that could answer the question but they did not show the results which seems odd to me.... "Data not shown". Therefore, they show these results in the new publication in comparison with those of amoxicillin.

Response: Thanks for your valuable comments. We did not originally use meropenem challenge simply on the basis that meropenem (or any carbapenem is not used in poultry farming). However, in accordance with your comments, we repeated the transfer in the presence of meropenem and show that it, similar to amoxicillin, increased the transfer efficiency of *bla*_{NDM-5}-IncX3. We have now added these data and describe that both meropenem and amoxicillin enhance the transfer efficiency of *bla*_{NDM-5}-IncX3 plasmid (Fig. 1 in rebuttal letter and Fig. S5 in the re-submission).

Fig. 1 The in vitro conjugation transfer efficiency of 3R strains carrying the IncX3 plasmid under meropenem and amoxicillin pressure.

Concerning your advice about the *in-vivo* study on the effect of meropenem on the transfer of *bla*_{NDM-5}-IncX3 plasmid, our application for doing this in chicken was rejected by our Institute Animal Experiment Ethics Committee, stating that the Chinese Ministry of Agriculture and Rural Affairs (MARA) prohibited the human critically-important antimicrobial agents (meropenem in this case) to be used in food-producing animals (chicken in this case). To address this limitation, we accessed and identified the enhancement of meropenem on the *in-vitro* conjugative transfer efficiency of *bla*_{NDM-5}-IncX3 plasmid (Fig. 1). Furthermore, hitherto, our *in-vivo* models have been chosen to mimic what antibiotics are currently used in global farming and whilst it may seem obvious that carbapenems may select NDM-5, it is less obvious that this selection pressure/modulation should be shown by an aminopenicillin.

Reviewer #4 (Remarks to the Author):

Reviewers' comments (remarks to the author)

In the reply to the Editor's comments authors included four points summarizing the aspects of novelty in their study. And these aspects are also included and discussed in the revised version of the manuscript.

- "The veterinary used β -lactam antibiotic amoxicillin can enhance the conjugation and persistence of *bla*_{NDM-5}-IncX3-carrying *E. coli* in the chicken gut".

Amoxicillin does not enhance conjugation. The word "enhance" should be changed with "select". The animal model formally demonstrates that amoxicillin is sufficient to maintain the IncX3- *bla*_{NDM-5} population, and there is no need to use carbapenems. This result is expected since the *bla*_{NDM-5} has a large spectrum against beta-lactams. Therefore, amoxicillin can sustain the colonization of the NDM-5 producers. Furthermore, the presence of the *bla*_{CTX-M-65} resistance gene on the chromosomes of 3R and 8R strains also plays its role in sustaining the growth of these strains in amoxicillin-treated animals. In this context, the discussion and conclusions about the conjugation and stability of the IncX3 vehicle obtained in the chicken gut model comparing amoxicillin treated animals with untreated animals should consider the positive selection on amoxicillin-resistant strains.

Response: Thank you for your constructive comments. We changed the word "enhance" to "select" in our revision in lines 295 and 301. We agree, and have shown, that amoxicillin alone is sufficient to sustain the colonization of the NDM-5 producers, as it possesses a broad catalytic spectrum against β -lactams. Following your concerns of chromosome-borne *bla*_{CTX-M-65} may play a role in maintaining 3R or 8R in amoxicillin-treated animals, we deleted and replaced *bla*_{CTX-M-65} in 3R with chloramphenicol resistant gene *cat*, and revealed that amoxicillin can increase the plasmid transfer efficiency to the similar level (0.003 ± 0.001 vs. 0.005 ± 0.002 , $p=0.1215$) in either 3R or 8R without *bla*_{CTX-M-65} (Fig. 2), indicating that chromosome-borne *bla*_{CTX-M-65} gene only plays a minor role in the conjugative transfer of *bla*_{NDM-5}-IncX3 plasmid under the presence of amoxicillin. This explanation has also been added in our re-submission as Figure S5.

Fig. 2 The in vitro conjugation transfer efficiency of 3R and 3R Δ bla_{CTX-M-65} strains carrying the IncX3 plasmid under amoxicillin pressure.

It is not clear if experiments in panels E and F of Figure 1 were performed in presence or absence of amoxicillin.

Response: The experiments in panels E and F were performed in the absence of amoxicillin, which has been added in the figure legends in lines 425-426.

-IncX3-carrying *E. coli* can persist in the chicken gut in the absence of antibiotic, and the novel IncX3 plasmid-borne *virBR* gene was identified to be responsible for the gut persistence of *bla*_{NDM-5}-IncX3-carrying *E. coli*".

The work adds novel information on identification and regulation of the T4SS of the IncX3 plasmids. Previous studies on adherent-invasive *E. coli* demonstrated that T4SS plays a role in the gut colonization [29]. The role of pili production produced under *VirBR* regulation should be more clearly stated, otherwise it seems that the entire conjugative process is linked to persistence of the *E. coli*.

Response: Many thanks for your valuable suggestion, we agree that the role of pili production should be more clearly stated. Accordingly, we have added the role of pili production in the adhesion and invasion not only in the discussion (lines 326-327 and lines 341-348), but also in the abstract (line 32-33).

-The heavy metal feed-additives, such as Cu and Zn ions play a key role as an exogenous signal in plasmid conjugative transfer by regulating *VirBR*-associated T4SS, therefore promoting the persistence of IncX3 plasmid-carrying *E. coli* in the absence of antibiotic. This dataset is in addition to the previous submission".

These conclusions are based on in vitro conjugation efficiency and mRNA measurements. There is no evidence that these metals interfere with the transcription. At least there is not sufficient strong demonstration to include this aspect in the title of the article. The Cu and Zn

effect was not tested in vivo. I strongly suggest including this evidence in the text but not in the title, it is not the major outcome of the study. It is overstated.

Response: We agree with your comment on this point and revised our title back to the first version accordingly. Here are the revised title: *VirBR*, a novel transcription regulator, promotes IncX3 plasmid conjugation, spread, and persistence of NDM-5 in animals.

Reviewer#1 comments

1. “We will explore in detail how heavy metal ions regulate *virBR* gene expression in further studies as our current work mainly focused on the identification, characterization and function of *VirBR*”. The heavy metal ions regulation has not been explored. In the revised version the activation by those exogenous signals is presented as mRNA abundance of *VirBR* and T4SS in the presence or absence of ZnSO₄ or CuSO₄. This is a preliminary result despite it is an interesting observation.

Response: We agree with your further comments that this is indeed a interesting observation, and we are exploring the mechanisms by which heavy metals promote plasmid conjugative transfer to explain this observation. The mechanism of ZnSO₄ or CuSO₄ is ongoing and will be part of an additional follow-up article.

2. The response of the authors is acceptable. They provided several additional experiments including *actX* promoter experiments and His6-*VirBR* assays and better EMSA experiments. However, the experiment with BW25113 demonstrates that it should be added conjugation efficiency could be “donor strain dependent”.

Response: Thank you very much for your supplementary response. Indeed, due to the different donor strains (BW25113 and 3R), even with the same recipient strain, there might be some variations in conjugation efficiency. However, the overall trend remains largely consistent.

3. The response to the third, fourth and fifth major concerns are acceptable.

Response: Thank you for your positive comments.

4. Could *VirBR* promote persistence not through the demonstrated effects on *tra* gene expression but by regulating another plasmid- or chromosomally encoded fitness trait?

Response: Please see our response to your third major comment listed above.

The pili role in adhesion should be better specified.

Response: Thank you very much for your guidance. We supplemented the content regarding the pili role in both discussion (lines 326-327 and lines 341-348) and abstract (line 32-33).

Reviewer #2 comments

- The reply “the veterinary used β -lactam antibiotic amoxicillin can enhance the conjugation and persistence of *bla*_{NDM-5}-IncX3-carrying *E. coli* in the chicken gut” is not acceptable. The most important problem here is that this sentence suggests that the authors are describing the amoxicillin as a positive regulator of the conjugative cluster, and as inducer of persistence. The *E. coli* cells carrying the plasmid in these experiments are resistant to amoxicillin, while the endogenous gut flora is not. Therefore, a higher levels (pink boxes) in Fig 1.b) are expected by

the positive selection of the 3R strain. Which is the experiment demonstrating that amoxicillin enhances in vitro the conjugation efficiency of the 3R donor?

Response: We appreciate your valuable comments. We changed the word “enhance” to “select” according to your suggestion in reviewer 4’s comments, and we apologise for not providing *bla*_{NDM-5}-IncX3 plasmid conjugative transfer data under pressure of amoxicillin in our previous response. The added *in-vitro* experiments with amoxicillin indicated a significant increase in *bla*_{NDM-5}-IncX3 conjugative transfer efficiency in the presence of amoxicillin (Fig. 3 in rebuttal letter and Fig. Sxx in re-submission).

Fig. 3 The in vitro conjugation transfer efficiency of 3R strains carrying the IncX3 plasmid under meropenem and amoxicillin pressure.

-The reply to the question “Which is the other b-lactam resistance genes associated with NDM-5 gene on the same Inc plasmid?” should be extended with a consideration that the 3R and 8R strains carried a *bla*_{CTX-M} gene in the chromosome.

Response: We sincerely apologize for the misunderstanding of the reviewer's question. the chromosome of 3R carried three antibiotic resistance genes, including *bla*_{CTX-M-65}, *tet*(B), *fosA*. In addition, the chromosome of 8R carried six antibiotic resistance genes, including *bla*_{CTX-M-65}, *tet*(B), *dfrA12*, *aadA8b*, *fosA*, *aph*(4)-Ia.

REVIEWERS' COMMENTS

Reviewer #3 (Remarks to the Author):

I thank very much the authors for having performed the experiment I suggested and understand that it could be performed in vitro only for ethical reasons. It is a very important and new result because it shows that amoxicillin, an antibiotic of very wide use (and on the access list of WHO) is even more potent than carbapenem to promote carbapenem resistance genes. Thus it implies that restriction of carbapenem use might not be sufficient at all to limit carbapenem resistance if amoxicillin use is not restricted also. This could have major implications on antibiotic policies. For that reason I think that a sentence on this result and these implications should be included in the discussion of the paper and also in the summary to give it maximum exposition.

Reviewer #4 (Remarks to the Author):

I appreciated the modifications done in the revised version following suggestions and comments.

REVIEWER COMMENTS

Reviewer #3 (Remarks to the Author):

I thank very much the authors for having performed the experiment I suggested and understand that it could be performed in vitro only for ethical reasons. It is a very important and new result because it shows that amoxicillin, an antibiotic of very wide use (and on the access list of WHO) is even more potent than carbapenem to promote carbapenem resistance genes. Thus it implies that restriction of carbapenem use might not be sufficient at all to limit carbapenem resistance if amoxicillin use is not restricted also. This could have major implications on antibiotic policies. For that reason I think that a sentence on this result and these implications should be included in the discussion of the paper and also in the summary to give it maximum exposition.

Response: Thank you very much for your positive feedback and suggestions regarding our research. Following your advice, we have emphasized in the Discussion of the manuscript that merely prohibiting the use of carbapenems in livestock farming is insufficient to combat carbapenem-resistant bacteria (Line 373-376).

Reviewer #4 (Remarks to the Author):

I appreciated the modifications done in the revised version following suggestions and comments.

Response: Thank you for your feedback. I'm glad to hear that you appreciated the modifications made in the revised version following your suggestions and comments.